# Identification and Extraction Optimization of Active Constituents in *Citrus junos* Seib ex TANAKA Peel and Its Biological Evaluation

**DOI:** 10.3390/molecules24040680

**Published:** 2019-02-14

**Authors:** Jung-hyun Shim, Jung-il Chae, Seung-sik Cho

**Affiliations:** 1Department of Pharmacy, College of Pharmacy, Mokpo National University, Muan, Jeonnam 58554, Korea; s1004jh@gmail.com; 2Department of Dental Pharmacology, School of Dentistry and Institute of Oral Bioscience, BK21 Plus, Chonbuk National University, Jeonju 56443, Korea

**Keywords:** *Citrus junos* Seib ex TANAKA, antioxidant, xanthine oxidase, elastase

## Abstract

*Citrus junos* Seib ex TANAKA possesses various biological effects. It has been used in oriental remedies for blood circulation and the common cold. Recently, biological effects of *C. junos* peel have been reported. However, optimization of the biological properties of *C. junos* peel preparations has yet to be reported on. We developed a high-performance liquid chromatography (HPLC) method for quantification of the active constituents in *C. junos* peel. Hot water and ethanolic extracts of *C. junos* peel were prepared and their chemical profiles and biological activities were evaluated. The 80% ethanolic extract demonstrated the greatest antioxidant activity and phenolic content, while the 100% ethanolic extract had the greatest xanthine oxidase inhibitory activity. Elastase inhibition activity was superior in aqueous and 20% ethanolic extracts. The contents of two flavonoids were highest in the 100% ethanolic extract. We postulated that the antioxidant and anti-aging effects of *C. junos* peel extract could be attributed to phenolics such as flavonoids. Our results suggest that the flavonoid-rich extract of *C. junos* may be utilized for the treatment and prevention of metabolic disease and hyperuricemia while the water-soluble extract of *C. junos* could be used as a source for its anti-aging properties.

## 1. Introduction

*Citrus junos* Seib ex TANAKA, a species of the family Rutaceae, is native to the southern coast and Jeju Island in Korea and China [1]. *C. junos* has been used in traditional medicine, cosmetics, and edible foods [2,3,4]. *C. junos* fruit has been traditionally used to improve blood circulation and treat the common cold [5]. It has been reported that *C. junos* contains many bioactive compounds such as vitamins, flavonoids, and limonoids that show anti-inflammatory and/or antioxidant activities [6]. The extract of *C. junos* can inhibit platelet aggregation, prevent ventricular dysfunction, and exert an antidiabetic effect [1,5,6]. Its fruits have been used as tea and its peels have been used as a source of essential oil. The *C. junos* peel has been dried and used as a raw material for tea. Recently, the biological effects of *C. junos* peel have been reported. Nakajima et al. have reported that *Citrus junos* peel can attenuate dextran sulfate sodium-induced murine experimental colitis and that its anti-inflammatory effect is related to its bioactive components such as hesperidin and naringin [7]. Shin et al. have found that 70% ethanolic extract of *C. junos* peel can reduce oleic acid-induced hepatic lipid accumulation in HepG2 cells with hypocholesterolemic effect in high-cholesterol diet mice models [8]. However, Shin et al. did not give reasons for or show active markers about why 70% ethanol extract was used in their experiment. Kim et al. have also reported the anti-diabetic effect of *C. junos* extract and its biomarkers such as rutin, hesperidin, quercetin [5].

On the other hand, there are no reports about optimization or the biological properties of *C. junos* peel extracts. Thus, the objective of this study was to investigate the active compounds and the biological activities of *C. junos* peel extracts for the development of natural medicine and as a source for cosmetics. Extraction optimization and standard analytical methods for quality control in plant sources utilization were important steps. Isolation and separation techniques were used to aid the identification of plant sources [9].

However, there is no standard profile for *C. junos* peel. Thus, in the present study, we established the quality control method using HPLC to separate and quantify hesperidin and naringin. We also investigated the optimum extraction of *C. junos* peel and the biological activities of these extracts. The optimized extract from *C. junos* peel was prepared and evaluated for its antioxidant, xanthine inhibitory, and elastase inhibitory activities in vitro.

## 2. Results and Discussion

### 2.1. Antioxidant Activity of C. junos Peel Extracts

Antioxidant potentials of hot water and ethanolic extracts of *C. junos* peels were determined by measuring 2,2-diphenyl-1-picrylhydrazyl (DPPH) scavenging activity, reducing power, and total phenolics. DPPH scavenging assay is a simple method for evaluating the free radical scavenging capacity of *C. junos* peel extracts. The antioxidant activities of natural sources are closely related to their phenolic components. Phenolic-rich sources from plant materials with antioxidant activity have diverse benefits against conditions such as oxidative imbalance and other metabolic diseases [9]. Thus, the antioxidant capacity of *C. junos* peel will provide important basic data for the development of medicinal and cosmetic materials.

Measured DPPH radical scavenging activity is shown in Table 1. The 80% ethanol extract showed the highest DPPH radical scavenging activity (IC_50_: 1042.37 μg/mL). A low IC_50_ value indicates strong antioxidant activity of a sample. The scavenging effects based on IC_50_ data of DPPH radicals were in the following order: 80% EtOH extract (1042.37 μg/mL) > 60% ethanol extract (1226.76 µg/mL) > 40% ethanol extract (1329.41 µg/mL) > 100% ethanol extract (1754.14 µg/mL) > hot water extract (2160.89 µg/mL) > 20% ethanol extract (2560.64 µg/mL).

We tested the reducing power of various extracts. The reducing power assay is a method of measuring the reducing power of an extract using ferrous ions. The 80% ethanolic extract exhibited the highest activity among all extracts (Table 2). The reductive activity expressed as vitamin C equivalents was 24.99 ± 0.35 μg/100 μg ex as extract. Reducing ability expressed as vitamin C equivalents was in the order: 80% ethanol extract (24.99 ± 0.35 μg/100 μg ex) > 60% ethanol extract (23.32 ± 0.27 μg/100 μg ex) > 40% ethanol extract (22.90 ± 0.28 μg/100 μg ex) > 20% ethanol extract (22.21 ± 0.46 μg/100 μg ex) > 100% ethanol extract (21.75 ± 0.38 μg/100 μg ex) > hot water extract (18.22 ± 0.20 μg/100 μg ex).

Total phenol content is tested using the reaction of Folin–Ciocalteu solution and phenolic compound [10]. Results are reported as gallic acid equivalents by referencing to a standard curve (*r*^2^ > 0.999) as shown in Table 2. Phenolic content in 80% ethanolic extract was higher than that in other extracts (25.44 ± 0.46 mg/g as gallic acid equivalents). Taken together, these results indicate that DPPH radical scavenging activity, reducing power, and phenolic contents were significantly higher in the 80% ethanolic extract than in other extracts.

In our previous report, we found that 80% ethanol was a more efficient solvent in the extraction of phenolic compounds from *C. coreana*. However, the contents of four main markers such as bergenin, quercetin, quercitrin, and isosalipurposide were increased when they were extracted with ethanol [9]. Thus, we concluded that phenolic extraction might be affected by solvent combinations. In the present study, 80% ethanolic extract showed the most excellent antioxidant activities.

### 2.2. Xanthine Oxidase Inhibitory Activity of C. junos Peel Extracts

Xanthine oxidase inhibitory activities of various solvent extracts of *C. junos* peel are shown in Figure 1. Allopurinol (Allo) at a concentration of 50 μg/mL significantly inhibited xanthine oxidase activity (99.75%). The xanthine oxidase inhibitory activity of the 100% ethanolic extract was significantly higher than that of other extracts at a concentration of 1 mg/mL (55.74%). Previously, we have reported that various botanical extracts are potential xanthine oxidase inhibitors [11]. Yoon et al. [12,13] have found that extracts of *Corylopsis coreana* and *Camellia japonica* inhibit xanthine oxidase activity (by 50%) at a concentration of 2 mg/mL. Yoon et al. [11] have reported that *Quercus acuta* extract shows approximately 50% xanthine oxidase inhibition at a concentration of 1 mg/mL. Optimized extract *Cudrania tricuspidata* showed xanthine oxidase inhibition by approximately 75% at a concentration of 2 mg/mL [9]. Activities of 100% ethanolic extract were two times stronger than extracts of *Corylopsis coreana* and *Camellia japonica*, but similar to *Quercus acuta* extract. Plant extracts with xanthine oxidase inhibitory activity at 1 and 2 mg/mL demonstrated consistent effects in a hyperuricemic mouse model. Thus, it is plausible that the 100% ethanolic extract of *C. junos* peel could be developed as a candidate anti-gout (anti-hyperuricemic) agent.

### 2.3. Elastase Inhibitory Activity of C. junos Peel Extracts

Elastase is a protease enzyme that degrades elastin. Inhibition of elastase can prevent skin aging [14]. An anti-elastase assay was performed to determine the ability of phytochemicals to inhibit elastase activity. Elastase inhibitory activities of various solvent extracts of *C. junos* peels are shown in Figure 2. Phosphoramidon (PPRM, positive control) at a concentration of 0.5 mg/mL significantly inhibited elastase activity (57.6 ± 8.33%). Elastase inhibitory activities of 20% ethanolic extract and hot water extract were significantly higher than those of other extracts at concentration of 1.0 mg/mL (61.4 ± 0.26% and 56.3 ± 0.6%, respectively).

Elastase inhibition of *C. junos* peel extract showed an opposite trend of antioxidant ability. Antioxidant activity was higher for ethanol extracts whereas elastase inhibition activity was higher for aqueous extracts. Elastase inhibitors are associated with anti-wrinkle and anti-aging properties. They can be developed into cosmetic materials [14]. An assessment of the anti-elastase activity of a plant extract can be a useful indicator of its potential application in cosmetic agents. Hot water and 20% ethanolic extract of *C. junos* peel exhibited more anti-elastase activity compared to other extracts (56.3 and 61%). However, this anti-elastase activity of the hot water extract could not be attributed to the presence of phenolics as reported in previous studies showing that phenolics such as flavonoids and tannins exhibited significant elastase inhibitory properties [15,16]. Therefore, studies on elastase inhibitory compounds of *C. junos* peel need to be conducted in the future.

### 2.4. Optimization of the Chromatographic Conditions and Contents of Marker Compounds from C. junos Extracts

In the present study, we investigated the analysis condition to separate two flavonoids such as naringin and hesperidin. A gradient program was used to separate the naringin and hesperidin in a single run within a practical period of time (Table 3). Chromatograms of standards and sample solutions are shown in Figure 3.

Information on retention time (RT) and quantification range, limit of detection (LOD) and limitation of quantification (LOQ) are summarized in Table 4. Limitation of detection of an individual analytical procedure is the lowest amount of an analyte in a sample that can be detected but not necessarily quantified. The limitation of quantification of an individual analytical procedure is the lowest amount of analyte in a sample that can be determined with suitable precision and accuracy. LODs for naringin and hesperidin were found to be 0.78 and 6.09 μg/mL, respectively. LOQ values for naringin and hesperidin were found to be 2.57 and 20.11 μg/mL, respectively (Table 4).

In previous reports, several extracts of *C. junos* peel were analyzed and several compounds were confirmed to be present in *C. junos* peel as marker compounds [4]. This finding is of significance in the use of this plant for industrial purposes. Two main peaks were identified as hesperidin and naringin in chromatographic profiles of extracts of *C. junos* peel. Kim et al. have previously reported that *C. junos* peel contains hesperidin and naringin, in agreement with our results [5]. We compared contents of these two active compounds in extracts of *C. junos* peel. Contents of these two compounds were found to be the highest in the 100% ethanolic extract (Table 5). The contents of hesperidin and naringin in the 100% ethanolic extract were 7.48 ± 0.04% and 0.63 ± 0.002%, respectively.

Hesperidin is a well-known flavonoid with anti-oxidant [17], anti-inflammatory [18], and immune modulatory activities [19]. Recently, Lee et al. have reported that hesperidin can inhibit UVB-induced increase of skin thickness, wrinkle formation, and collagen fiber loss in male hairless mice. Hesperidin significantly inhibited the increase of epidermal thickness and epidermal hypertrophy and suppressed expression levels of MMP-9 and pro-inflammatory cytokines. These results indicate that hesperidin has anti-photoaging activity in UVB-irradiated hairless mice [20]. Contents of hesperidin in *C. junos* peel extract was 4.67 to 7.48%. Kim et al. have reported that hesperidin and naringin concentrations in 70% extract of *C. junos* peel are 0.03% and 0.01% (*w/w*), respectively [5]. In the present study, flavonoid-rich extracts containing hesperidin and naringin (7.48% and 0.63%, *w/w* respectively) were obtained.

Naringin (flavanone-7-*O*-glycoside) is known to have antioxidative, neuroprotective, and anti-inflammatory activities [21,22,23]. Naringin is also thought to contribute to anti-aging effects of the skin. Ren et al. have found that naringin can effectively protect skin against UVB-induced keratinocyte apoptosis and damage via inhibition of ROS (reactive oxygen species) production and COX-2 overexpression [24]. Additionally, Candhare et al. have reported that naringin ointment exerts wound healing potential via down-regulating expression levels of inflammatory factors, factors, and up-regulating expression of growth factors [25].

In previous reports on naringin and hesperidin, the DPPH radical scavenging activity of naringin was reported to be over 100 μg/mL [26]. Monica et al. described that naringin reduced Fe in a concentration dependent manner from 5 mM to 0.5 mM/mL to 15 mM at 0.1 mg/mL [27].

Srimathi et al. reported the antioxidant effects of hesperidin. IC_50_ of DPPH radical scavenging activity of hesperidin was 41.55 μg/mL. The reducing power of hesperidin was found to be 47.46 μg/mL while the standard antioxidant ascorbic acid was of 35.35 μg/mL [28]. The amount of hesperidin in the 80% extract was calculated to be 33.54 μg/100 μg extract eq based on the reducing power prescribed by Srimathi et al. However, the hesperidin content of the 80% extract was calculated to be 7.48% (7.48 μg/100 μg extract *w/w*), and actually, hesperidin contributes to the reducing power of the 20% portion. Therefore, naringin and hesperidin were not considered to be the major influencing factors for the antioxidant ability of *C. junos* peel extract. Thus, naringin and hesperidin are markers of *C. junos* peel, but various other antioxidants should be identified as marker/active compounds.

Both hesperidin and naringin are thought to be suitable as anti-oxidant and anti-inflammatory markers, as well as anti-aging markers of *C. junos* peel extract. *C. junos* peel is considered suitable as anti-aging material because all extract samples show elastase inhibitory activity. As shown in Figure 2, the elastase inhibitory activity was the highest in the hot water extract and 20% ethanolic extract. Besides, hesperidin and naringin as good antioxidant and anti-inflammatory agents showed the highest contents in 100% ethanol extract. Therefore, hesperidin and naringin are anti-aging markers in *C. junos* peel extract. Other water-soluble elastase inhibitors might be present in the hot water extract and 20% ethanolic extract. Thus, studies on elastase inhibitory compounds of water-soluble fraction from *C. junos* peel need to be conducted in the future.

Taken together, these results suggest that 80% ethanolic extract is suitable as an antioxidant while 100% ethanolic extract is suitable as an xanthine oxidase inhibitor. Meanwhile, hot water extract and 20% ethanolic extract are suitable as cosmetic materials showing an anti-aging effect.

None of previous studies reported so far have stated diverse activities and constituents of extracts of *C. junos* peel. To the best of our knowledge, our present study is the first to report the optimization of the extraction process of pharmaceutically and/or cosmetically active indicators from *C. junos* peel and to compare the antioxidant, xanthine oxidase inhibitory, and elastase activities of various extracts of *C. junos* peel.

## 3. Materials and Methods

### 3.1. Plant Material and Extract Preparation

*C. junos* peel was supplied from Korea Yuju Inc. (Gwangju, Korea). A voucher specimen (MNUCSS-CJ-01) was deposited at the Mokpo National University (Muan, Korea). Peels were dried and extracted for the present study. These air-dried and powdered *C. junos* peels (10 g) were extracted with 20%, 40%, 60%, 80%, and 100% ethanol (100 mL) at room temperature for 3 days. The 0% extract was prepared using hot water extraction (100 °C, 4 h). After filtration, the residual part was evaporated, freeze-dried and then stored at −70 °C before analysis.

### 3.2. DPPH Free Radical Assay

Antioxidant activity of the sample was determined using 2,2-diphenyl-1-picrylhydrazyl (DPPH) radical scavenging assay. Briefly, sample solution (1 mL) containing extract was added to DPPH sample solution (0.4 mM, 1 mL) and mixed at room temperature for 10 min. Absorbance was recorded at 517 nm using a microplate reader (Perkin Elmer, Waltham, MA, USA). DPPH free radical scavenging activities of samples were compared based on their IC_50_ (μg/mL) values [10].

### 3.3. Reducing Power

The reducing power of extracts was determined following a modified reducing power assay method. The sample (0.1 mL) was mixed with sodium phosphate buffer (0.2 M, 0.5 mL) and potassium ferricyanide (1%, 0.5 mL) followed by incubation at 50 °C for 20 min. Subsequently, trichloroacetic acid solution (10%, 0.5 mL) was added to the reaction mixture. Reaction mixture was centrifuged at 12,000 rpm for 10 min. The supernatant was mixed with distilled water (0.5 mL) and iron (III) chloride solution (0.1%, 0.1 mL). The absorbance was recorded at 700 nm. Reducing powers of extracts were expressed as vitamin C equivalents [10].

### 3.4. Determination of Total Phenolic Content

Folin–Ciocalteu assay was used for the quantifcation of the total phenolic content [10]. Sample or standard (1 mL) was mixed with sodium carbonate solution (2%, 1 mL) and Folin–Ciocalteu’s phenol reagent (10%, 1 mL) followed by incubation at room temperature for 10 min. Absorbance was measured at 750 nm using a microplate reader (Perkin Elmer, Waltham, MA, USA) and compared with a calibration curve of gallic acid. Results were expressed as milligrams of gallic acid equivalents per gram of sample [10].

### 3.5. Determination of Xanthine Oxidase Inhibitory Activity

Xanthine oxidase assay is a method to measure the amount of uric acid produced by xanthine oxidase as described previously [10]. Phosphate buffer (100 mM; pH 7.4, 0.6 mL), sample (0.1 mL), xanthine oxidase (0.2 U/mL, 0.1 mL), and xanthine (1 mM; dissolved in 0.1 N NaOH, 0.2 mL) was mixed at 37 °C for 30 min and HCl (1 N, 0.2 mL) was added to finish the reaction. Allopurinol was used as a positive control. The absorbance was measured at 290 nm using a microplate reader (Perkin Elmer, Waltham, MA, USA).

### 3.6. Determination of Elastase Inhibitory Activity

The elastase inhibitory assay was used with a slight modification of the method described by Chiocchio et al. [29]. Briefly, elastase from porcine pancreas (10 μg/mL, 10 μL) was mixed with Tris-HCl (0.2 M, 90 μL), STANA (2.5 mM, *N*-Succinyl-Ala-Ala-Ala-*p*-nitroanilide, 100 μL), and sample (50 μL) at 37 °C for 30 min. After completion of the reaction, the supernatant was centrifuged at 15,000 rpm for 10 min. The absorbance was measured at 405 nm using a microplate reader (Perkin Elmer, Waltham, MA, USA). Phosphoramidon was used as a positive control.

### 3.7. Chemical Profiling by HPLC Analysis

All analyses were performed using an Alliance 2695 HPLC system (Waters, Milford, MA, USA). A reverse phase C18 column (5-μm, 150 mm × 5 mm) was used with a mobile phase consisting of a mixture of solvent A (acetonitrile) and B (0.2% phosphoric acid). The gradient was programmed as follows: 0–45 min: 15% A; 45–50 min: a linear gradient from 15 to 100% A; 50–51 min: a linear gradient from 100 to 15%; 51–55 min: 15% A. The flow rate was 0.8 mL/min. The UV detector was set at 280 nm.

## 4. Conclusions

The present study reveals both that the 80% ethanolic extract of *C. junos* peel possesses antioxidant activity and that the 100% ethanolic extract possesses a xanthine oxidase inhibitory effect. Hot water extract and 20% ethanolic extract possess elastase inhibitory activities. In addition, it is hypothesized that the photochemicals present in the *C. junos* peel might be responsible for biological activities of its extracts. Results of this study provide an excellent foundation for future development of *C. junos* peel-based medicinal and/or cosmetic preparations.

## Figures and Tables

**Figure 1 molecules-24-00680-f001:**
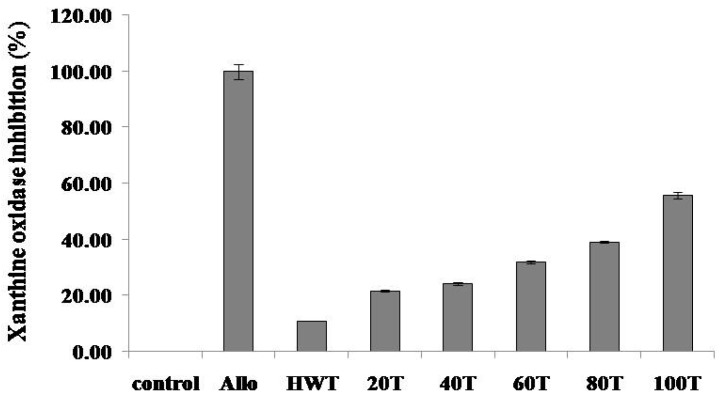
Xanthine oxidase inhibitory activities in extracts of *C. junos* peel (1 mg/mL) and allopurinol (Allo, 50 μg/mL). Allo; allopurinol, HWT; hot water, 20T; 20% EtOH ex, 40T; 40% EtOH ex, 60T; 60% EtOH ex, 80T; 80% EtOH ex, 100T; 100% EtOH ex, Values were the mean ± standard deviation (*n* = 3).

**Figure 2 molecules-24-00680-f002:**
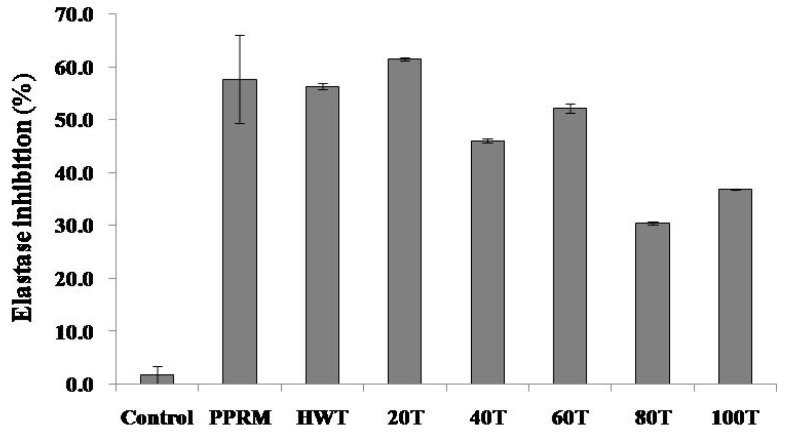
Elastase inhibitory activities in various solvent extracts of *C. junos* peel (1.0 mg/mL) and Phospharamidon (PPRM, 0.5 mg/mL), HWT; hot water, 20T; 20% EtOH ex, 40T; 40% EtOH ex, 60T; 60% EtOH ex, 80T; 80% EtOH ex, 100T; 100% EtOH ex, Values were the mean ± standard deviation (*n* = 3).

**Figure 3 molecules-24-00680-f003:**
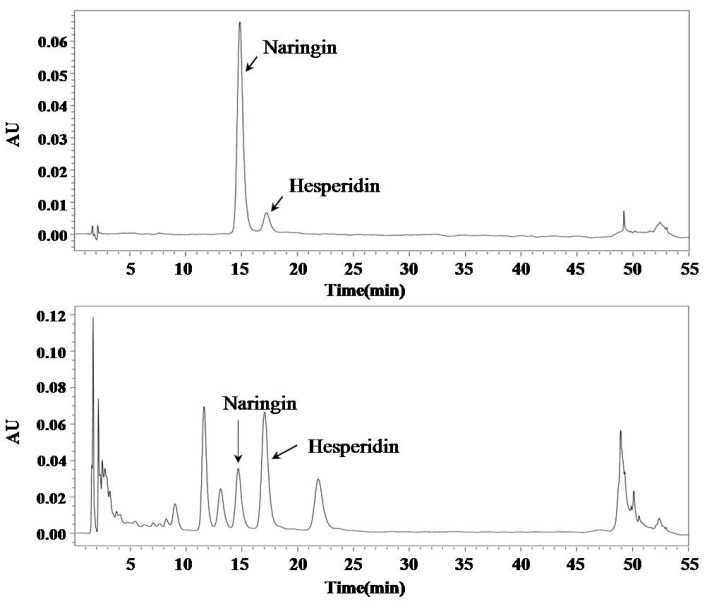
Bioactive constituent profiles of *C. junos* peel using HPLC.

**Table 1 molecules-24-00680-t001:** DPPH radical scavenging effect of extracts from *C. junos* peel (IC_50_ value).

	DPPH Scavenging Activity IC_50_ (μg/mL)
Vitamin C	8.09
Hot water	2160.89
20% EtOH ex	2560.64
40% EtOH ex	1329.41
60% EtOH ex	1226.76
80% EtOH ex	1042.37
100% EtOH ex	1754.14

**Table 2 molecules-24-00680-t002:** Reducing power and total phenolic contents of *C. junos* peel extracts.

Extract	Reducing Power (Ascorbic Acid eq. μg/100 μg Extract)	Total Phenolic Content (Gallic Acid eq. mg/g)
Hot water	18.22 ± 0.20	17.22 ± 0.27
20% EtOH ex	22.21 ± 0.46	20.43 ± 0.23
40% EtOH ex	22.90 ± 0.28	21.67 ± 0.4
60% EtOH ex	23.32 ± 0.27	22.81 ± 0.58
80% EtOH ex	24.99 ± 0.35	25.44 ± 0.46
100% EtOH ex	21.75 ± 0.38	24.77 ± 0.21

**Table 3 molecules-24-00680-t003:** Analytical conditions of HPLC system for analyzing markers.

Parameters	Conditions		
Column	Zorbax extended-C18		
	(C18, 4.6 mm × 150 mm, 5 µm)		
Flow rate	0.8 mL/min		
Injection volume	10 μL		
UV detection	280 nm		
Run time	55 min		
**Gradient**	**Time (min)**	**A (%)**	**B (%)**
	0	15	85
	45	15	85
	50	100	0
	51	15	85
	55	15	85

**Table 4 molecules-24-00680-t004:** HPLC data for the calibration graphs and limit of quantification of naringin and hesperidin.

Analyte	Retention Time (min)	*r* ^2^	Linear Range (μg/mL)	LOQ (μg/mL)	LOD (μg/mL)
Naringin	15.3	0.9999	6.25–100	2.57	0.78
Hesperidin	17.7	0.9999	6.25–100	20.11	6.09

**Table 5 molecules-24-00680-t005:** Contents of naringin and hesperidin from *C. junos* peel extracts.

	Naringin (%)	Hesperidin (%)
Water	0.35 ± 0.006	4.67 ± 0.01
20% EtOH	0.44 ± 0.02	5.15 ± 0.09
40% EtOG	0.47 ± 0.003	5.92 ± 0.01
60% EtOH	0.44 ± 0.01	5.64 ± 0.01
80% EtOH	0.55 ± 0.01	7.07 ± 0.04
100% EtOH	0.63 ± 0.002	7.48 ± 0.04

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
