# Peer review of "Identification and Extraction Optimization of Active Constituents in Citrus junos Seib ex TANAKA Peel and Its Biological Evaluation"

_molecules, 2019, doi:10.3390/molecules24040680_

Round 1
Reviewer 1 Report
Opinion related to the paper entitled „Identification and extraction optimization of active constituents in Citrus junos Seib ex TANAKA peel and its biological evaluation”.
The paper is interesting, especially the inhibition of elastase and xanthine oxidase by the extracts. My own old research (patent) has shown that the grapefruit peel extract ointment was very effective in inhibiting skin lesions in rats with induced porphyria. This was probably related to the alleviation of oxidative stress (singlet oxygen). However, in simple tests DPPH, ABTS, FRAP reducing properties of compounds such as hesperidin or naringin are small in comparison to flavones such as luteolin or flavonols such as quercetin. In my opinion, the results demonstrated by Authors do not prove the therapeutic importance of hesperidin or naringin as antioxidants. The beneficial effects of hesperidin or naringenin in oxidative stress (in vivo) could be related to the action of these compounds on various metabolic pathways (enzymes). If Authors believe that hesperidin and naringin are strong antioxidants (with significant reducing properties due to their polyphenolic structure) should support this by citing more works showing (in simple tests in vitro) antioxidant properties of these compounds.
Other remarks:
lines 40 and 41 should be hesperidin and naringin;
line 43 insert the space after model;
line 54 should be naringin;
line 55 text needs an English correction;
line 61 sentence is imprecise, needs correction;
lines 70, 72 1,042 data 1,042 the data incompatible with Table 1;
line 94 add respectively after μg/mL;
line 107 should be C. junos peel
line 112 should be Corylopsis coreana, Camellia japonica;
line 115 Cudrania tricuspidata;
line 117 should be Corylopsis coreana and Camellia japonica, … Quercus acuta;
line 120 should be C. junos peel;
line 129 better to write “…determine the ability of phytochemicals to inhibition of elastase activity”;
line 131 why Figure 2 is written in red;
line 137 the sentence seems incomplete;
line 153 should be naringin……………….naringin;
line 172 space after word “results”;
line 178 insert space after words antioxidant and anti-inflammatory;
line 219 such a long heating of extracts may cause degradation of active compounds;
line 234 sextracts?
Author Response
Response to the Reviewers’ Comments-1
Opinion related to the paper entitled „Identification and extraction optimization of active constituents in Citrus junos Seib ex TANAKA peel and its biological evaluation”.
The paper is interesting, especially the inhibition of elastase and xanthine oxidase by the extracts. My own old research (patent) has shown that the grapefruit peel extract ointment was very effective in inhibiting skin lesions in rats with induced porphyria. This was probably related to the alleviation of oxidative stress (singlet oxygen). However, in simple tests DPPH, ABTS, FRAP reducing properties of compounds such as hesperidin or naringin are small in comparison to flavones such as luteolin or flavonols such as quercetin. In my opinion, the results demonstrated by Authors do not prove the therapeutic importance of hesperidin or naringin as antioxidants. The beneficial effects of hesperidin or naringenin in oxidative stress (in vivo) could be related to the action of these compounds on various metabolic pathways (enzymes). If Authors believe that hesperidin and naringin are strong antioxidants (with significant reducing properties due to their polyphenolic structure) should support this by citing more works showing (in simple tests in vitro) antioxidant properties of these compounds.
Response: We thank the reviewers insightful comment. The important point of our study is that the utilization of C. junos depends on the extraction method. Various biological activity depend on the extraction condition. The hesperidin and naringin mentioned in this manuscript are widely known as the components of C. junos. Analysis and evidence are already widely known. We presented our analysis method for the validity of the analysis. The analysis data are limited to the data presented in order to determine the relation with biological activity.
The point in this study is that the amount of phenolics, antioxidant activities and elastase activities of the extracts are not in direct proportion. We also described that hesperidin, naringin is an indicator but not an main elastase inhibitor. And studies on these characteristics should be made in the future, and are clearly described in the discussion section.
Other remarks:
lines 40 and 41 should be hesperidin and naringin;
Response: According reviewer’s kind recommendation, the revised texts in the manuscript are highlighted with red color.
line 43 insert the space after model;
Response: According reviewer’s kind recommendation, the revised texts in the manuscript are highlighted with red color.
line 54 should be naringin;
Response: According reviewer’s kind recommendation, the revised texts in the manuscript are highlighted with red color.
line 55 text needs an English correction;
Response: According reviewer’s kind recommendation, the revised texts in the manuscript are highlighted with red color.
line 61 sentence is imprecise, needs correction;
Response: According reviewer’s kind recommendation, the revised texts in the manuscript are highlighted with red color.
lines 70, 72 1,042 data 1,042 the data incompatible with Table 1;
Response: According reviewer’s kind recommendation, the revised texts in the manuscript are highlighted with red color.
line 94 add respectively after μg/mL;
We thank the reviewers insightful comment. We removed some sentences to correct the context according to reviewers comments. Deleted sentences are marked with strikethrough.
line 107 should be C. junos peel
Response: According reviewer’s kind recommendation, the revised texts in the manuscript are highlighted with red color.
line 112 should be Corylopsis coreana, Camellia japonica;
Response: According reviewer’s kind recommendation, the revised texts in the manuscript are highlighted with red color.
line 115 Cudrania tricuspidata;
Response: According reviewer’s kind recommendation, the revised texts in the manuscript are highlighted with red color.
line 117 should be Corylopsis coreana and Camellia japonica, … Quercus acuta;
Response: According reviewer’s kind recommendation, the revised texts in the manuscript are highlighted with red color.
line 120 should be C. junos peel;
Response: According reviewer’s kind recommendation, the revised texts in the manuscript are highlighted with red color.
line 129 better to write “…determine the ability of phytochemicals to inhibition of elastase activity”;
Response: According reviewer’s kind recommendation, the revised texts in the manuscript are highlighted with red color.
line 131 why Figure 2 is written in red;
Response: According reviewer’s kind recommendation, the revised texts in the manuscript are highlighted with black color.
line 137 the sentence seems incomplete;
Response: According reviewer’s kind recommendation, the revised texts in the manuscript are highlighted with red color.
line 153 should be naringin……………….naringin;
Response: According reviewer’s kind recommendation, the revised texts in the manuscript are highlighted with red color.
line 172 space after word “results”;
Response: According reviewer’s kind recommendation, the revised texts in the manuscript are highlighted with red color.
line 178 insert space after words antioxidant and anti-inflammatory;
Response: According reviewer’s kind recommendation, the revised texts in the manuscript are highlighted with red color.
line 219 such a long heating of extracts may cause degradation of active compounds;
Response: I agree with the reviewer's comments. The hot water extraction conditions (100C, 4 hr) are the experimental conditions that we usually use and we did not consider materials that are unstable to heat.
line 234 sextracts?
Response: According reviewer’s kind recommendation, the revised texts in the manuscript are highlighted with red color.
Reviewer 2 Report
The paper deals with a high performance liquid chromatography (HPLC) method used for quantification of active constituents in C. junos peel. The chemical profiles and biological activities were evaluated for hot water and ethanolic extracts of C. junos peel. In particular antioxidant activity, phenolic content, xanthine oxidase and elastase inhibition activities were evaluated for several extracts obtained using different water/ethanol ratios. Unfortunately only two flavonoids were quantified in the different extracts and apparently only characterized by their retention time in the chromatographic runs. (No MS spectra no other analytical data but the chromatogram for the standards are reported). In my opinion for a publication in a journal such as Molecules at least the major peaks obtained in the chromatograms of the different performed extractions should be identified (MS or other available assessment method) and quantified. This is extremely important in order to correlate the observed biological effects with the molecular content of the extracts. The chemical structures of the identified compounds (including but not limited to the two flavonoids narigin and hesperidin) should also be reported. This should be an easy task for the authors taking into account that, as they refer in their manuscript, most of the compounds contained in C. junos peel have been already identified (refs 4 and 5).
Author Response
Response to the Reviewers’ Comments-2
The paper deals with a high performance liquid chromatography (HPLC) method used for quantification of active constituents in C. junos peel. The chemical profiles and biological activities were evaluated for hot water and ethanolic extracts of C. junos peel. In particular antioxidant activity, phenolic content, xanthine oxidase and elastase inhibition activities were evaluated for several extracts obtained using different water/ethanol ratios. Unfortunately only two flavonoids were quantified in the different extracts and apparently only characterized by their retention time in the chromatographic runs. (No MS spectra no other analytical data but the chromatogram for the standards are reported). In my opinion for a publication in a journal such as Molecules at least the major peaks obtained in the chromatograms of the different performed extractions should be identified (MS or other available assessment method) and quantified. This is extremely important in order to correlate the observed biological effects with the molecular content of the extracts. The chemical structures of the identified compounds (including but not limited to the two flavonoids narigin and hesperidin) should also be reported. This should be an easy task for the authors taking into account that, as they refer in their manuscript, most of the compounds contained in C. junos peel have been already identified (refs 4 and 5).
Response: We thank the reviewers insightful comment.
Points of our experiments is that various biological activity depend on the extraction condition. The hesperidin and naringin mentioned in this manuscript are widely known as the components of C. junos. In our study, we have found that hesperidin and naringin are present in large amounts in the C. junos. Additively, analysis and evidence are already widely known. Therefore, we did not provide additional analysis data for marker compounds.
We presented our analysis method for the validity of the analysis using HPLC. The reason for using the HPLC analysis method is that the marker compound is already well known. If HPLC analysis is difficult because hesperidin and naringin are similar in structure or molecular weight or isomer, we will use HPLC-MS or other alternative column or tools.
In addition, we thought that industrial analysis of food/medicinal materials and by-products should be as easy and fast as HPLC methods. We also thought that schools and companies should have the convenience of using the equipment like HPLC.
The results of our analysis are presented in figure 3 and table 3. The analysis data are limited to the data presented in order to determine the relation with biological activity.
The second point in this study is that the amount of phenolics, antioxidant activities and elastase activities of the extracts are not in direct proportion. We also described that hesperidin, naringin is an indicator but not an main elastase inhibitor. And studies on these characteristics should be made in the future, and are clearly described in the discussion section.
Reviewer 3 Report
To me, the scope of the paper could be larger. Optimization of extraction is not present. Only, different proportions of ethanol and water (at room temperature) and 100% hot water were used as solvents.
The only novelty is showing that 80% ethanolic extract could be suitable as an antioxidant, 100% ethanolic extract could be suitable as an xanthine oxidase inhibitor while hot water extract and 20% ethanolic extract could be suitable as anti-aging cosmetics.
However, the highest contents of the marker compounds (hesperidin and naringin) were determined when 100% ethanol was used. To me, discussion in this area is not sufficient.
Some statements are not clear to me:
“However, this anti-elastase activity of hot water extract could not be attributed to the presence of phenolics as reported in previous studies showing that phenolics such as flavonoids and tannins exhibited significant elastase inhibitory properties[19,20].”
Some parts are poorly written:
“The C. junos peel has been dried and used as a raw material for tea. Recently, effects on the C. junos peel have been reported. C. junos peel has been dried and used as a raw material for tea. Recently, biological effects of C. junos peel have been reported.”
or
“They reported that efficacy of caffeic acid derivatives of extraction was increased when it was..”
Some parts are not strictly in the subject:
“Similarly, we have previously reported that the extraction efficiency of caffeic acid from pear pomace is related to the water soluble solvent[13]. Besides, the extraction efficiency of chlorogenic acid from pear pomace is related to ethanol ratio in extraction solvent [22]. These results are also related to solubilities of active compounds.”
or
“Activities of 100% ethanolic extract were two times stronger than extracts of Corylopsis coreana and Camellia japonica, but similar to Quercus acuta extract. Plant extracts with xanthine oxidase inhibitory activity at 1 and 2 mg/mL demonstrated consistent effects in hyperuricemic mouse model.”
Author Response
Response to the Reviewers’ Comments-3
Question 1
To me, the scope of the paper could be larger. Optimization of extraction is not present. Only, different proportions of ethanol and water (at room temperature) and 100% hot water were used as solvents. The only novelty is showing that 80% ethanolic extract could be suitable as an antioxidant, 100% ethanolic extract could be suitable as an xanthine oxidase inhibitor while hot water extract and 20% ethanolic extract could be suitable as anti-aging cosmetics. However, the highest contents of the marker compounds (hesperidin and naringin) were determined when 100% ethanol was used. To me, discussion in this area is not sufficient. Some statements are not clear to me:“However, this anti-elastase activity of hot water extract could not be attributed to the presence of phenolics as reported in previous studies showing that phenolics such as flavonoids and tannins exhibited significant elastase inhibitory properties[19,20].”
Response: We thank the reviewers insightful comment. Hesperidin and naringin have been reported in previous papers as markers of Junos peel. The important point in this study is that the amount of phenolics and elastase activities of the extracts are not in direct proportion. We also described that hesperidin, naringin is an indicator but not an main elastase inhibitor. And studies on these characteristics should be made in the future, and are clearly described in the discussion section.
-->Thus, both hesperidin and naringin are thought to be suitable as anti-oxidant and anti-inflammatory markers as well as anti-aging markers of C. junos peel extract. It is considered that C. junos peel is suitable as an anti-aging material because all extract samples show elastase inhibitory activity. As shown in Figure 2, the elastase inhibitory activity was the highest in the hot water extract and 20% ethanolic extract. Besides, hesperidin and naringin as good antioxidant and anti-inflammatory agents showed the highest contents in 100% ethanol extract. Therefore, hesperidin and naringin are anti-aging markers in C. junos peel extract, other water-soluble elastase inhibitors might be present in the hot water extract and 20% ethanolic extract. Thus, studies on elastase inhibitory compounds of water soluble fraction from C. junos peel need to be conducted in the future.
Question 2
Some parts are poorly written: “The C. junos peel has been dried and used as a raw material for tea. Recently, effects on the C. junos peel have been reported. C. junos peel has been dried and used as a raw material for tea. Recently, biological effects of C. junos peel have been reported.”
Response: We thank the reviewers insightful comment. We deleted the incorrectly written sentence.
“They reported that efficacy of caffeic acid derivatives of extraction was increased when it was..”
Response: We thank the reviewers insightful comment. Hole sentence is as follows.
-> They reported that efficacy of caffeic acid derivatives of extraction was increased when it was extracted with hot water (116 μg/mL).
Question 3
Some parts are not strictly in the subject:
“Similarly, we have previously reported that the extraction efficiency of caffeic acid from pear pomace is related to the water soluble solvent[13]. Besides, the extraction efficiency of chlorogenic acid from pear pomace is related to ethanol ratio in extraction solvent [22]. These results are also related to solubilities of active compounds.”
Response: We thank the reviewers insightful comment. We removed some sentences to correct the context according to reviewers comments. Deleted sentences are marked with strikethrough.
Kim et al.have prepared hot water, 50%, and 100% ethanolic extracts from Ligularia fischeri. They reported that efficacy of caffeic acid derivatives of extraction was increased when it was extracted with hot water (116 μg/mL). In 50% and 100% ethanolic extracts, contents of caffeic acid derivatives were 72 and 11 μg/mL[12]. Caffeic acid derivatives are thought to be highly extracted by hot water due to their solubility. Similarly, we have previously reported that the extraction efficiency of caffeic acid from pear pomace is related to the water soluble solvent[13]. Besides, the extraction efficiency of chlorogenic acid from pear pomace is related to ethanol ratio in extraction solvent [22]. These results are also related to solubilities of active compounds.
Question 4
“Activities of 100% ethanolic extract were two times stronger than extracts of Corylopsis coreana and Camellia japonica, but similar to Quercus acuta extract. Plant extracts with xanthine oxidase inhibitory activity at 1 and 2 mg/mL demonstrated consistent effects in hyperuricemic mouse model.”
Response: We thank the reviewers insightful comment. We have described the XO efficacy of other plant extracts to compare C. junos peel to other materials. In order for readers to understand the possibility of C. junos as anti-hyperuricemic source, I thought it was necessary to describe the XO activity of other plant extracts.
Reviewer 4 Report
This work is well structured. The innovation lies in the identification and quantification of hesperidin and narigin.
Two minor corrections are needed.
Table 1. Not μg/ml but μg/mL
Table 5. Not 10 μl but 10 μL
Author Response
Response to the Reviewers’ Comments-4
This work is well structured. The innovation lies in the identification and quantification of hesperidin and narigin.
Two minor corrections are needed.
Table 1. Not μg/ml but μg/mL
Table 5. Not 10 μl but 10 μL
Response: We thank the reviewers insightful comment. we changed the typo in Table 1 and 5. The revised texts in the manuscript are highlighted with red color
Round 2
Reviewer 2 Report
The authors declined to deal with referee suggestions. The only partially identified active compound in the paper are Naringin and Hesperidin (both summing up to less than 8% of the extracts) but the observed biological effects are ascribable to the different extracts containing plenty of several unknown and not quantified metabolites.
In order to make more sound their conclusion:
"The present study reveals that 80% ethanolic extract of C. junos peel possesses antioxidant activity and 100% ethanolic extract possesses xanthine oxidase inhibitory effect. Hot water extract and 20% ethanolic extract possess elastase inhibitory activities. In addition, it is hypothesized that photochemicals present in the C. junos peel might be responsible for biological activities of its extracts. Results of this study provide an excellent foundation for future development of C. junos peel based medicinal and/or cosmetic preparations."
The authors should at least compare the results of extracts biological effects with the biological effects of pure Naringin and Hesperidin mixtures having concentrations comparable to those found in their extracts.
Author Response
Response to the Reviewers’ Comments-2
The authors declined to deal with referee suggestions. The only partially identified active compound in the paper are Naringin and Hesperidin (both summing up to less than 8% of the extracts) but the observed biological effects are ascribable to the different extracts containing plenty of several unknown and not quantified metabolites.
In order to make more sound their conclusion:
"The present study reveals that 80% ethanolic extract of C. junos peel possesses antioxidant activity and 100% ethanolic extract possesses xanthine oxidase inhibitory effect. Hot water extract and 20% ethanolic extract possess elastase inhibitory activities. In addition, it is hypothesized that photochemicals present in the C. junos peel might be responsible for biological activities of its extracts. Results of this study provide an excellent foundation for future development of C. junos peel based medicinal and/or cosmetic preparations."
The authors should at least compare the results of extracts biological effects with the biological effects of pure Naringin and Hesperidin mixtures having concentrations comparable to those found in their extracts.
Response: We thank the reviewers insightful comment. We have found the privious reports on naringin and hesperidin. DPPH radical scavenging activity of naringin was reported to be over 100 μg/mL. Monica et al described that naringin reduced Fe in a concentration dependent manner from 5 mM to 0.5 mM/mL to 15 mM at 0.1 mg/mL.
Srimathi et al reported the antioxidant effect of hesperidin. IC50 of DPPH radical scavenging activity of hesperidin was 41.55 μg/mL. The reducing power of hesperidin was found to be 47.46 μg/mL while the standard antioxidant ascorbic acid was of 35.35 μg/mL. The amount of hesperidin in the 80% extract was calculated to be 33.54 ug/100 ug extract eq based on the reducing power rescribed by Srimathi et al. However, the hesperidin content of 80% extract was calculated to be 7.48% (7.48 ug/100ug extract w/w), and actually, hesperidin contributes to the reducing power of 20% portion. Therefore, naringin and hesperidin were not considered to be the major influencing factors for the antioxidant ability of C. junos peel extract. Thus, naringin and hesepridin are markers of C. junos peel, but various other antioxidants should be identified as marker/active compounds. We described in the discussion section in red color.
Reviewer 3 Report
Some of my suggestions were taken into account and quality of the paper increased.
Some linguistic errors still exist, e.g.:
Line 41
???have found that 70% ethanolic extract of C. junos peel can reduce oleic 41 acid-induced hepatic lipid accumulation in HepG2 cells with hypocholesterolemic effect in high-cholesterol diet mice model [8].
Line 150
In the presnet study, we investigated the analysis condition to separate two falvonoids such as...
Author Response
Response to the Reviewers’ Comments-3
Some of my suggestions were taken into account and quality of the paper increased.
Some linguistic errors still exist, e.g.:
Line 41
???have found that 70% ethanolic extract of C. junos peel can reduce oleic 41 acid-induced hepatic lipid accumulation in HepG2 cells with hypocholesterolemic effect in high-cholesterol diet mice model [8].
Response: We thank the reviewers insightful comment. we corrected the context with red color
Line 150
In the presnet study, we investigated the analysis condition to separate two falvonoids such as...
Response: We thank the reviewers insightful comment. we corrected the typo with red color